# Seismic evidence for melt-rich lithosphere-asthenosphere boundary beneath young slab at Cascadia

Xin Wang [1,2,3], Ling Chen [2,4] ✉, Kelin Wang[5], Qi-Fu Chen [1,2], Zhongwen Zhan[3] & Jianfeng Yang[2,4]

The Lithosphere-Asthenosphere Boundary (LAB) beneath oceanic plates is generally imaged as a sharp seismic velocity reduction, suggesting the presence of partial melts. However, the fate of a melt-rich LAB is unclear after these plates descend into the mantle at subduction zones. Recent geophysical studies suggest its persistence with down-going old and cold slabs, but whether or not it is commonly present remains unclear, especially for young and warm slabs such as in the Cascadia subduction zone. Here we provide evidence for its presence at Cascadia in the form of a large ($9.8 \pm 1.5\%$) decrease in shear-wave velocity over a very small (<3 km) depth interval. Similarly large and sharp seismic velocity reduction at the bottom of both old and young slabs, as well as along the base of oceanic plates before subduction, possibly represents widespread presence of melts. The melt-rich sub-slab LAB may strongly influence subduction dynamics and viscoelastic earthquake cycles.

The Lithosphere-Asthenosphere Boundary (LAB) beneath oceanic plates marks a sharp decrease in seismic velocity. The sharp decrease, loosely but commonly referred to as a negative "seismic discontinuity", is widely thought to indicate the presence of partial melts[1–5], although there are competing interpretations[6,7]. A melt-rich LAB may mechanically decouple the lithospheric plate from the underlying asthenosphere and thus facilitate plate motion[8,9]. If it continues into subduction zones, similar decoupling would occur beneath the subducting plate (called the slab) and fundamentally influence subduction dynamics[10] and viscoelastic earthquake cycles[11]. Here we refer to the LAB of an oceanic plate before subduction as the plate-LAB and that after subduction as the slab-LAB (Fig. 1). However, it is by no means clear whether the slab-LAB is as common as the plate-LAB. The slab-LAB has been well imaged in the Japan and New Zealand subduction zones[1,8], and in both cases, the age of the subducting plate is very old[12] (~120–130 Ma; Fig. 1a). In contrast, its presence in regions where young plates are subducting remains unclear[3]. A magneto-telluric study off the Middle America trench could resolve a melt-rich

plate-LAB beneath the young (~23 Ma) Cocos plate but not a slab-LAB in the subduction zone[13]. The plate-LAB is seismically detected beneath almost the entire young (<10 Ma) Juan de Fuca plate, but so far, there is no reported evidence for the slab-LAB in the Cascadia subduction zone[14]. Whether or not the presence of a melt-rich slab-LAB depends on slab age is a question with important geodynamic implications.

Cascadia is an end-member warm-slab subduction zone where the incoming Juan de Fuca plate is young (<10 Ma) and warm (Fig. 1). Using teleseismic converted waves generated at seismic discontinuities (referred to as receiver functions, RFs), previous studies reported various discontinuities related to the subducting Juan de Fuca slab, including those associated with the fluid-rich subducting oceanic crust, slab-Moho, and anisotropy within the slab crust and/or mantle[15–22]. However, no seismic discontinuity associated with slab-LAB has been reported thus far. Seaward of the subduction zone, Rychert et al. detected[14] the plate-LAB throughout the incoming Juan de Fuca plate and considered it strong evidence for the presence of partial melts along the base of the plate. However, whether the inferred

[1]Key Laboratory of Earth and Planetary Physics, Institute of Geology and Geophysics, Chinese Academy of Sciences, Beijing, China. [2]College of Earth and Planetary Sciences, University of Chinese Academy of Sciences, Beijing, China. [3]Seismological Laboratory, California Institute of Technology, Pasadena, CA, USA. [4]State Key Laboratory of Lithospheric Evolution, Institute of Geology and Geophysics, Chinese Academy of Sciences, Beijing, China. [5]Pacific Geoscience Centre, Geological Survey of Canada, Sidney, BC, Canada. ✉ e-mail: lchen@mail.iggcas.ac.cn

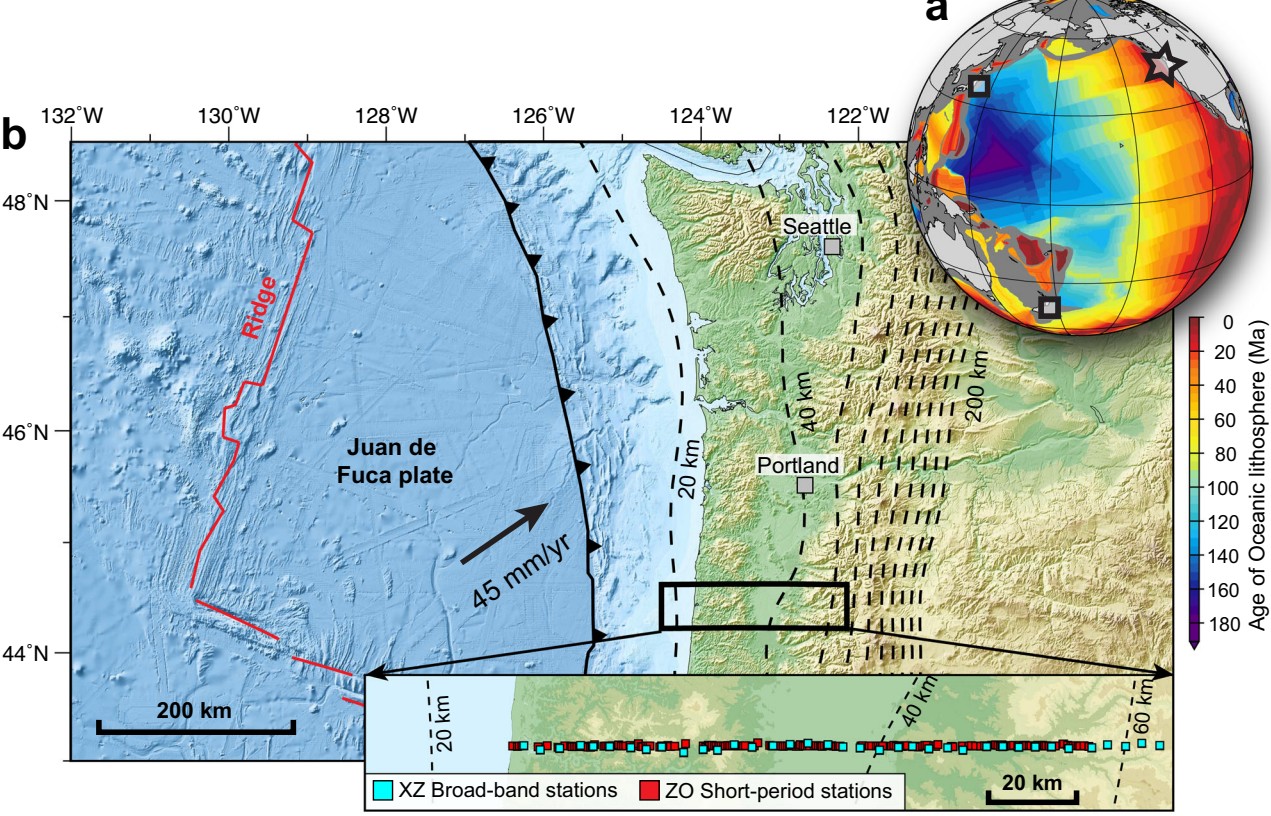

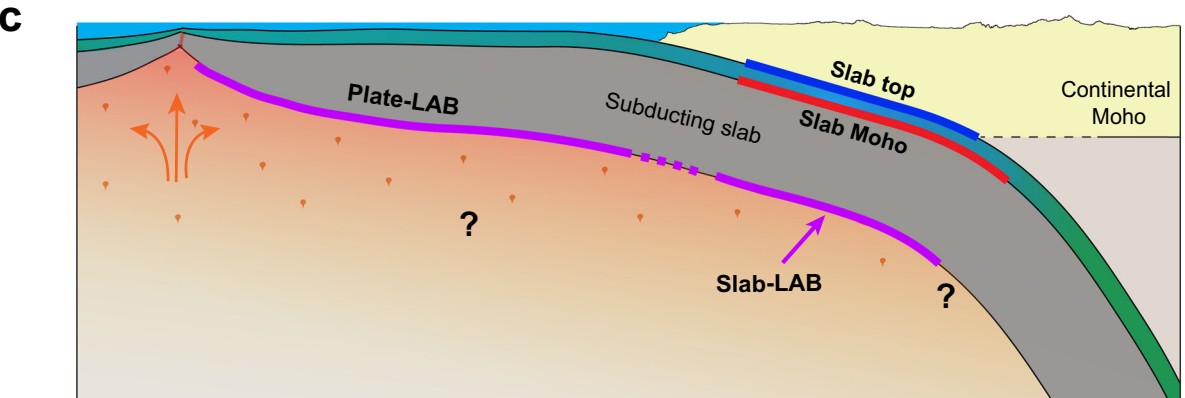

**Fig. 1 | Study area and illustration of the seismic discontinuities discussed in this study. a** Global map showing the age distribution of the oceanic lithosphere with data from ref. 12. Our study area (star), the Cascadia subduction zone, has a young and warm slab, in contrast to the old and cold slab in the other two sub-duction zones (squares; Japan and New Zealand) where evidence for sub-slab melt-rich Lithosphere-Asthenosphere Boundary (LAB) has been reported[1,8]. **b** Regional tectonic setting of our study area and locations of seismic stations used in this study. Dashed lines show depth to the surface of the slab from Slab2.0 model[59]. The maps were generated using Generic Mapping Tools[60], with topography and bathymetry data from the Global Multi-Resolution Topography Synthesis[61]. **c** Cartoon showing how the LAB beneath the Juan de Fuca plate (plate-LAB, based on ref. 14) may continue after subduction to give rise to the slab-LAB. Question marks indicate that the origination of melts for the LAB remains under debate, and the depth extent of the slab-LAB requires further investigation.

melt-rich LAB persists after the plate is subducted remains an unre-solved question.

To address this question, we examine P-to-S (Ps conversion) RFs by taking advantage of dense seismic arrays at Cascadia (Fig. 1). One array consists of broadband seismometers with a station spacing of ~5 km and operated in 1993–1994; the other consists of short-period nodal seismometers with a station spacing of ~500 m and operated for one month in 2017 (Supplementary Fig. 1). Data from these arrays have been used in previous studies for investigating the shallow sedimen-tary basin structures, mid-crustal Conrad discontinuity, continental

Moho, slab geometry, fluid-rich slab crust and its dehydration pro-cesses, and fossil anisotropy within the slab mantle[15,17,20–26]. Below the slab-Moho, some patchy negative RF signals have been detected previously[21,22,24,25], which might be indicative of the presence of a slab-LAB. However, these studies did not consider them to be structural signals because of their intermittent appearance and concerns about artifacts arising from the side-lobe effect of RF deconvolution and multiple reverberations from shallow structures. One study[21] tentatively attributed these negative signals to anisotropy within a very thick slab, yet their findings might be influenced by cross-mode

artifacts resulting from the usage of a multiple-mode conversion imaging technique[20]. In our present study, by employing a recently developed Bayesian array-based coherent receiver function (CRF) imaging technique[27,28], and conducting multiple frequency analysis and extensive synthetic tests, we are able to detect the slab-LAB from these data in the form of strong and sharp seismic velocity reduction. We consider the imaged slab-LAB to be compelling evidence for the continuation of the melt-rich plate-LAB after subduction.

## Results

### CRF imaging and comparison with conventional RF analyses

In the distance range (30°–90°) optimal for CRF analyses, the arrays recorded seismic waves generated by distant earthquakes mainly in the southeast, northwest, and southwest directions (Supplementary Fig. 1). Because the Juan de Fuca slab dips to the east, the Ps waves from the southeast are more effective than those from the other directions in constraining slab-parallel seismic discontinuities (Supplementary Figs. 2, 3). We, therefore, mainly use the Ps waves from this direction for our analyses. The results based on waves from the other directions provide corroborative information. We process the data in multiple frequency bands (0.05–0.5 Hz, 0.15–1.0 Hz, 0.25–1.5 Hz, 0.35–2.0 Hz) and employ the CRF technique to conduct a joint analysis of the broadband and short-period data (see Methods; Supplementary Figs. 4–6). The denser spatial coverage of the short-period array allows us to better constrain shallow structures, which aids in constraining deep structures (see Methods).

Compared to previous RF studies[15,22,26], which predominantly focused on relatively low-frequencies (up to 0.3 Hz), we use multiple frequency analyses mentioned above to check the persistence of discontinuities. The frequency dependence of the amplitudes of Ps phases helps to appraise the robustness and sharpness of the discontinuities[29,30]. Although increasing the frequency content of the data increases the vertical resolution of subsurface structures, high-frequency analysis is prone to noise and local scattering. To overcome this challenge, we employ the array-based CRF technique, which can leverage the coherency of the wavefield recorded by a dense array to suppress incoherent noise and local scattering (Fig. 2). The CRF method is also less susceptible to interference to the target discontinuity image from nearby strong discontinuities, as shown by the comparison of subsurface images obtained using the CRF and conventional RF methods (Fig. 2 and Supplementary Fig. 7). Another advantage of the CRF is that the reliability of imaged structures is measured using probability distribution, allowing objective assessment of structure identification and interpretation[28].

The CRF images are originally obtained in the time domain. To convert time-to-depth, we employ the seismic velocity model from the tomography study of ref. 31. Although the spatial resolution of this model is much higher than other available regional tomography models[32–34] (Supplementary Fig. 8), its velocity field is still inevitably damped by the smoothing scheme in tomographic inversion and thus distorts the depths of discontinuities in the CRF imaging (Supplementary Figs. 9, 10). As will be detailed in the following section, correction for this distortion is an integral part of the interpretation of our CRF results. The CRF image based on seismic waves from the southeast is shown in Fig. 3. Seismic waves from the northwest are theoretically less optimal for resolving east-dipping discontinuities (Supplementary Fig. 2), but they still result in comparable CRF images for an area immediately to the north (Supplementary Fig. 11), offering further support to the main imaged features shown in Fig. 3. Because of the potential depth distortion discussed above, we refer to the depths shown in Fig. 3 as apparent depths.

### The slab-top and slab-Moho discontinuities

Shallower than 60 km, our CRF image reproduces several primary structure boundaries found in previous studies[16,20,21,25,35] and thus demonstrates the efficacy of the CRF technique. At 0–10 km apparent depths, prominent positive discontinuities (Fig. 3a) roughly delineate known sedimentary basins[26,35]. The negative intra-crustal discontinuity observed at about 10–20 km depth has also been observed by previous studies[20,25] and may be related to the accumulation of fluids within the continental crust[36,37]. In the 20–50 km range, two subparallel east-dipping seismic discontinuities with opposite polarity delineate the slab geometry and the low-velocity subducting crust. By utilizing higher-frequency data in comparison to previous studies[15,24], our proposed slab geometry contains more details; however, it is important to note that the tomographic model used in time-to-depth conversion may introduce slight distortions to the slab geometry, both in our study and in previous ones. The apparent thickness of the low-velocity zone >10 km is larger than what is expected for a normal oceanic crust due to the aforementioned depth distortion caused by velocity smoothing in the tomography model that is used for time-depth conversion. If we follow previous studies in Cascadia to assume a fluid-saturated crust with a very low Vs (about 20–50% Vs reduction) and a high Vp/Vs ratio (about 2–3)[16,17], the thickness is corrected to be ~7–8 km (Supplementary Fig. 10). The CRF amplitudes and our synthetic tests indicate extremely large velocity contrasts across the top (−15–20%) and bottom (−35–55%) of the subducting crust (see details in Methods, Fig. 4), consistent with previous estimates[16,17]. The decrease in the amplitudes of the Ps conversions at the slab-top and slab-LAB when the slab depth exceeds ~40 km (Fig. 3a) is likely due to the eclogitization process of the subducting oceanic crust[15,23]. Eclogitization is accompanied by dehydration and thus releases aqueous fluid into the overlying forearc mantle wedge, leading to the well-known absence of a detectable continental Moho near the mantle wedge corner[15].

### The slab-LAB discontinuity

The most important finding enabled by the newly employed CRF technique is the continuous negative discontinuity at about 60–80 km apparent depths, which we interpret as the slab-LAB (Fig. 3). A direct estimate of the actual depth of the negative discontinuity using observed Ps arrival times show that it is located about 25 km below the slab-Moho or about 32 km below the slab-top (Supplementary Fig. 10). Patchy negative signals at relevant depths can be seen in some other seismic imaging studies at Cascadia[20,21,24,25], but it was not clear whether they were structural signals or imaging artifacts. Here, we are able to resolve confidently this discontinuity owing to a combination of multiple frequency analyses, the narrow back-azimuth range of earthquakes used in dipping discontinuity imaging, and the advantages of the CRF method. The Bayesian posterior probability distribution of the CRF phases shows a high confidence level for this discontinuity (Fig. 3b), and its presence is persistent across multiple frequency bands (Supplementary Figs. 5, 6).

Our synthetic tests verify that the strong negative signal marking this discontinuity cannot be an imaging artifact due to other structures (Supplementary Figs. 12–16). Basin reverberations occur at shallower depths, and multiples from the slab-top occur at much deeper depths (Supplementary Fig. 12). Multiples from a positive intra-crustal discontinuity (i.e., the Conrad discontinuity) at ~15 km depth in the overriding continental plate would exhibit a pair of positive and negative signals at a depth similar to the observed negative discontinuity (Supplementary Fig. 13). However, such a Conrad discontinuity is not observed in our CRF image (Fig. 3) and the conventional RF images (Supplementary Fig. 5). Although a negative intra-crustal discontinuity is imaged at a depth of 10–15 km in our study, it generates multiples with geometry inconsistent with the slab-LAB discontinuity (Supplementary Figs. 13, 14). More importantly, the RF images constructed with seismic waves from the northwest show a weaker negative signal than that obtained using the waves from the southeast (Supplementary Figs. 3, 15), and the arrival time of the

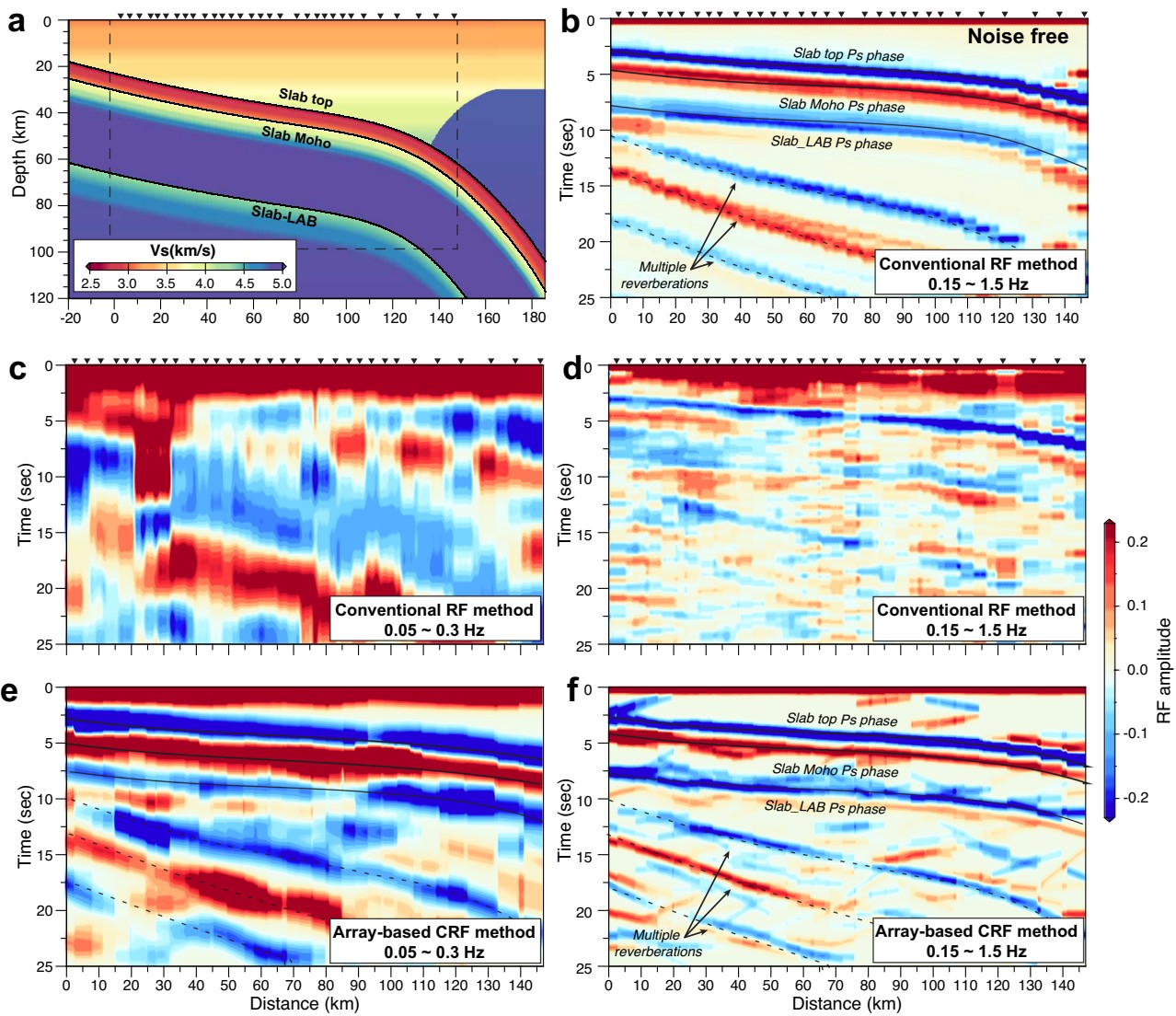

**Fig. 2 | Synthetic tests to compare the array-based coherent receiver function (CRF) method with the conventional receiver function (RF) method. a** The synthetic model based on the slab geometry in central Cascadia and the distribution of the broadband seismic stations (triangles) shown in Fig. 1b. Synthetic seismograms are generated for the specific source-receiver pairs (with the earthquakes located in the southeast direction in Supplementary Fig. 1) in central Cascadia to simulate real situations. **b** Conventional RF stacking images for noise-free synthetic data. In this scenario, only high-frequency (0.05–1.5 Hz) results are presented to highlight RF signals related to slab-related discontinuities. **c, d** Conventional RF stacking images for synthetic data with realistic noise. With low frequency (0.05–0.3 Hz), the slab-top and slab-Moho can be coherently identified only with their multiples, and it is difficult to recognize the slab-LAB. With high frequency (0.05–1.5 Hz), slab-related signals manifest as intermittent features. **e, f** Array-based CRF stacking images for synthetic data with realistic noise. The slab-top, slab-Moho, and slab-LAB can be resolved in both low- and high-frequency imaging. See Supplementary Fig. 7 for noise-free synthetic tests.

negative phase decreases with the increase of epicentral distance (Supplementary Fig. 16), suggesting that the negative signal is related to the direct Ps phase rather than multiples. We further analyze the RFs as a function of back-azimuth using both synthetic data and real observations, and our results show that this negative signal is better explained by a dipping discontinuity rather than an anisotropy layer (Supplementary Figs. 17–19).

**Sharpness and velocity reduction across the slab-LAB**
An important attribute of the inferred LAB discontinuity is its sharpness, characterized by the magnitude of the velocity change (dVp or dVs) and the depth range (dZ) over which the change occurs (Fig. 4). The amplitude of the Ps phase in RFs is particularly sensitive to dVs, dZ, and the range of dominant frequencies used in the analysis. If dZ is broader than half of the dominant wavelength of the incident P-wave, the amplitude decreases dramatically with increasing frequency[8,29]

(Supplementary Fig. 20). The Ps conversions of our inferred slab-LAB consistently exhibit an amplitude of $12 \pm 3\%$ of the direct P-arrival across the entire frequency range (Fig. 4). Assuming a typical upper-mantle Vp of ~8 km/s and considering the waveform frequencies used in this study up to 2.0 Hz, the dZ related to this discontinuity should not much exceed ~2 km[29]. Further synthetic tests indicate that the slab-LAB corresponds to a sharp discontinuity with dVs of $9.8 \pm 1.5\%$ over a depth range of $1.5 \pm 1.5$ km (Fig. 4 and Supplementary Fig. 21). Such a sharp velocity reduction within 3 km cannot be solely explained by variations in water abundance, temperature, or grain size[1,3,38]. A recent study[39] suggests that hydration within the asthenosphere can substantially reduce seismic velocities, offering a potential explanation for the strong and sharp velocity reduction across the LAB. However, the effect of hydration on velocity reduction becomes less pronounced for seismic waveform frequencies considered in this study. Thus, the presence of partial melts stands out as the most likely mechanism for

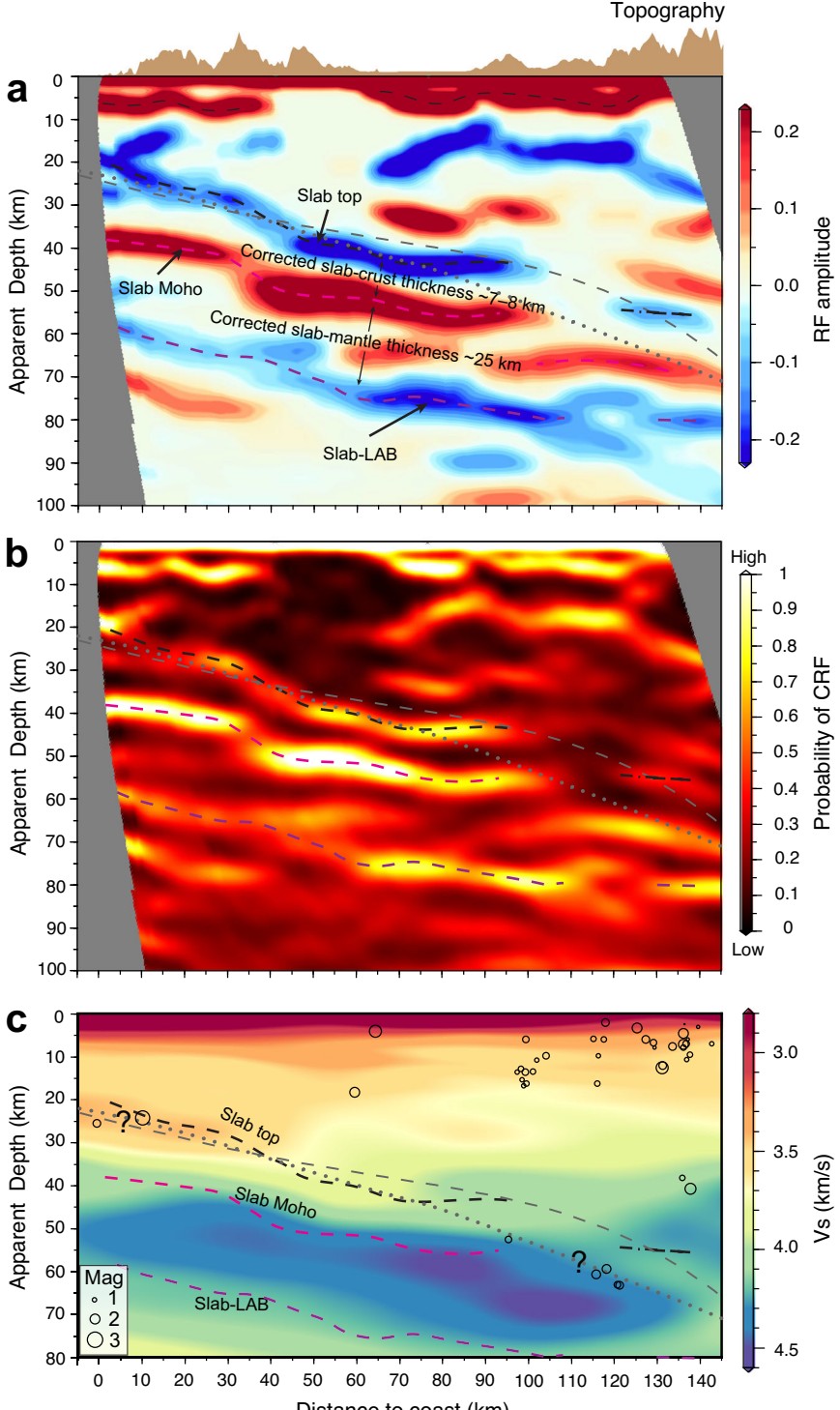

**Fig. 3 | Coherent receiver function (CRF) stacking image and interpretation.**
**a** CRF stacking image obtained by jointly inverting the broadband and short-period data using teleseismic waves (0.15–1.0 Hz) from the southeast. Interpretation of the three important discontinuities is as labeled. The apparent depths estimated for these discontinuities suffer from distortion caused by smoothing in the seismic velocity model employed for time-depth conversion. As discussed in the text and shown in Supplementary Fig. 10, the actual separations between the slab-top and slab-Moho and between the slab-Moho and slab-LAB are estimated to be ~7–8 and ~25 km, respectively. **b** Density plot of the ensemble solutions represents the posterior probability distribution of the CRF phases, which serves as a measure of the reliability of the imaged structures. **c** Same three CRF discontinuities from (a) in the backdrop of the tomographic image of ref. 31 used for our time-to-depth conversion. Open circles show earthquakes from the Pacific Northwest Seismic Network catalog within 10 km of the cross-section. In all the panels, the thin dashed and dotted gray lines represent the subduction interface from Slab2.0 model[59] and ref. 62, respectively.

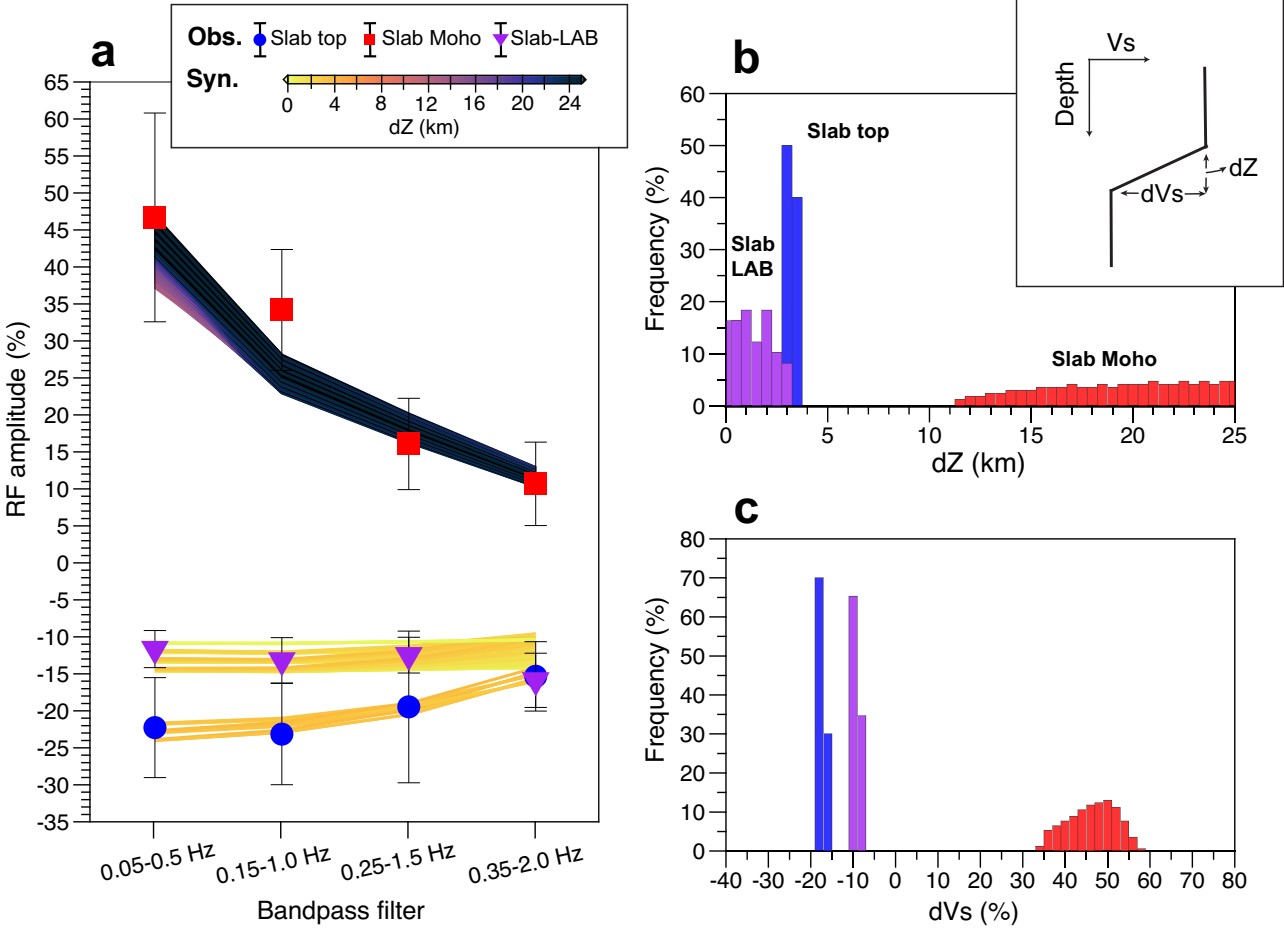

**Fig. 4 | Estimation of seismic velocity structures. a** Amplitude of the Ps phases as a function of frequency. The observations are shown with colored circles as defined in the legend, with the error bar representing one standard deviation. The synthetic counterparts are shown as lines color-coded by the depth range (dZ) over which velocity decrease/increase occurs, as defined in the legend. For display clarity, we only show the models with a relative misfit (as defined in Methods) of less than 100%. **b**, **c** Histograms showing dZ and the magnitude (dVs), respectively, of the velocity decrease/increase associated with the three discontinuities.

the strong and sharp velocity reduction at the slab-LAB. Similar to previous estimates[1,3,14], a dVs of ~9.8 ± 1.5% across the LAB can be attributed to the presence of ~1–4% melt fraction, depending on the melt geometry[40].

## Discussion

Negative seismic discontinuities below the slab-Moho have been previously recognized at Cascadia, but they are too shallowly (about 5–15 km) located beneath the slab-Moho to be the slab-LAB. They are thus inferred to indicate anisotropy of the subducting oceanic uppermost mantle from fossil fabrics generated at spreading ridges[18,21]. In contrast, our LAB discontinuity is located at a greater depth and outlines the lower boundary of the high-velocity slab (Fig. 3 and Supplementary Fig. 11). As explained above, detailed comparisons between real observations and synthetic data show that our imaged structure is more consistent with a dipping feature than with anisotropy. More importantly, its about 32 km depth below the top of the slab is compatible with the depth of the Juan de Fuca plate-LAB before subduction (Supplementary Fig. 22). It is thus natural to consider the slab-LAB imaged in this work a continuation of the plate-LAB.

It is widely assumed that partial melting at the uppermost asthenosphere is responsible for the presence of a melt-rich plate-LAB[3]. It is important to ask whether the partial melts at the slab-LAB are generated in situ with a similar origin or instead reflect the downdip

transport of plate-LAB melts by the subducting slab. Hawley et al. speculated[34] that the melt-rich plate-LAB material traveled laterally with the Juan de Fuca plate towards the Cascadia subduction zone but mainly piled up beneath the trench area because its buoyancy would resist subduction[13], giving rise to a tomographically detected large low-velocity volume. However, in a later tomographic study, Bodmer et al. did[41] not find the low-velocity volume in central Cascadia, including our study area, putting the notion of melt piling up and hence the downdip transport scenario in question. Nonetheless, neither of these tomographic studies has adequate vertical spatial resolution to resolve the sharpness of the LAB discontinuity imaged in our study.

Considering the inference of melt-rich LAB with comparable melt fractions beneath oceanic plates both before and after subduction[1,8,14], we speculate that the melts originate predominantly locally, without long-distance lateral migration from ridges or plumes. For the Juan de Fuca plate, Rychert et al. detected[14] a weaker and intermittent positive discontinuity at ~90–130 km depths near the mantle solidus as well as the plate-LAB at ~20–45 km depths. Partial melting may occur between the two discontinuities, with melt ponding at the base of the less-permeable oceanic lithosphere over geological timescales[42,43] to cause the observed sharp velocity reduction. A typical slab thermal model in Cascadia with a mildly hydrated mantle (water contents of 100–200 ppm) could also result in the occurrence of in situ partial melting to explain the observed sharp slab-LAB in our study (Supplementary

Fig. 23). However, a deeper (~100 km) positive seismic discontinuity near the mantle solidus is not imaged in our study. Whether or not its absence is due to its weaker Ps and/or interference from the multiples of the slab crustal discontinuities is a question for future research (Supplementary Figs. 12, 13).

The plate-LAB has been widely observed beneath the Pacific and Atlantic[3–5,44,45]. Though observations show an age dependence of the depths of the plate-LAB up to ~60 Ma[3,4], there is no clear age dependence of its degree of seismic velocity reduction (Supplementary Fig. 24). The slab-LAB beneath the ~10 Ma young, warm, and thin slab at Cascadia detected in this study is comparable to that beneath the ~130 Ma old, cold, and thick slab in the Japan subduction zone in terms of strength and sharpness[1]. The similarly large and sharp seismic velocity reduction at the bottom of both old and young slabs, as well as along the base of oceanic plates before subduction, suggests that a melt-rich LAB may occur regardless of slab age and may be commonly present in subduction zones. It may have evaded detection in most places due to difficulties in imaging dipping structures at mantle depths using conventional methods. The state of knowledge will become clearer as monitoring networks and imaging methodology improve. Experimental and theoretical studies show that the presence of melt reduces viscosity[46,47], and therefore the existence of a melt-rich LAB can influence subduction dynamics[8–10] and viscoelastic deformation in subduction earthquakes[11]. The decoupling effect of a melt-rich slab-LAB may necessitate a revision of subduction zone rheology at Cascadia and, therefore, a new look at its earthquake-cycle dynamics and megathrust locking[48,49].

## Methods

### Data

Seismic data used in this study are from two nearly collocated dense linear arrays in central Oregon, western US, including a broadband (BB) seismic array (with a flat response from ~0.01–50 Hz; https://www.fdsn.org/networks/detail/XZ_1993/) with a station spacing of ~5 km operated from 1993/03/23 to 1994/11/23 and a short-period (SP) nodal seismic array (with a corner frequency of 5 Hz; https://www.fdsn.org/networks/detail/ZO_2017/) with a station interval of ~500 m operated from 2017/06/23 to 2017/08/01 (Supplementary Fig. 1). Compared to the BB array, the SP array is ten times as dense but is limited by a narrower frequency band and a short observation duration. However, after removing the instrument response, frequencies greater than 0.1 Hz are generally well recovered by the SP nodal stations[25,28,50,51]. The longer temporal coverage of the BB data and the denser spatial coverage of the SP data complement each other in imaging subsurface structures, as explained in the following section.

### Array-based coherent receiver function imaging

We use a recently developed Bayesian array-based coherent receiver function (CRF) imaging technique for subsurface structure imaging[27,28]. The basic concept of the CRF is similar to that of the conventional single-station receiver function (RF) technique, but the CRF method includes array processing, in which the deconvolution at a single station involves constraints from nearby stations provided that the stations are closely located to detect similar subsurface structures sampled by seismic waves from multiple earthquakes in a narrow back-azimuth range. The basic assumption of the CRF method is that the teleseismic wavefields are coherent and the subsurface structures at nearby stations effectively share similar features. The dominant frequency of teleseismic P-waveforms is around 0.3–0.5 Hz with wavelengths longer than ~10–25 km, which ensures that signals recorded by the ~5 km distributed broadband stations show very similar waveforms, satisfying the coherent assumption required for CRF imaging. In addition, incorporating short-period data with a station spacing of approximately 500 m in CRF imaging improves the resolution of shallow structures, such as sedimentary basins. This, in turn, aids in constraining deeper structures

and enables us to conduct higher-frequency CRF analysis. Using CRF, we can suppress incoherent noise and local scattering to retrieve weak signals. Another advantage of the CRF technique is that the deconvolution is performed using a trans-dimensional Markov chain Monte Carlo (McMC) inversion algorithm[52,53]. Thus, the obtained RFs are presented in probability formats, enabling us to determine the reliability and uncertainties of the obtained RFs.

We first acquire waveform data from earthquakes with magnitude larger than 5.8 and epicentral distances from 30 to 90 degrees (Supplementary Fig. 1). The recorded seismic waves were generated by distant earthquakes mainly in the northwest, southeast, and southwest directions (Supplementary Fig. 1). In the presence of a dipping structure, RFs depend strongly on the direction of the incoming teleseismic waves[54] (Supplementary Figs. 2, 3). In our study area, the slab dips eastward. To optimize the imaging of the east-dipping slab-parallel seismic discontinuities[55], we mainly use waveforms from the southeast direction (earthquakes colored by orange in Supplementary Fig. 1) in the CRF imaging. The three-component seismograms are rotated into radial, transverse, and vertical components, windowed around the theoretical first P-arrivals, and filtered into four frequency bands (0.05–0.5 Hz, 0.15–1.0 Hz, 0.25–1.5 Hz, and 0.35–2.0 Hz). Compared to the low-frequency analyses (e.g., the 0.05–0.5 Hz used in this study and the up to 0.3 Hz used in previous studies), higher frequencies serve to reduce the wavelength of waveforms, thus increasing the vertical resolution of subsurface structure imaging. However, high-frequency analysis is easily affected by noise and local scattering. Nevertheless, the usage of the CRF method can better suppress incoherent noise and local scattering to improve subsurface structural imaging (Fig. 2), and the integration of multiple frequency data enables us to resolve the robustness and detail structures of the subsurface discontinuities. We then remove low-quality traces which have signal-to-noise ratios (SNR) less than 2 in the vertical component. Here SNR is defined as the ratio of the root mean squared amplitude over an 8-s window after the first P-arrival to that before the first P-arrival. After data preprocessing, 4073 teleseismic waveforms are retrained for the BB array, and 2138 waveforms are retrained for the SP array. The distribution of the P-to-S conversion points at a depth of 60 km, and the distribution of earthquakes in different back-azimuths are shown in Supplementary Fig. 1.

To image lithosphere-scale structure using CRF, we use a moving subarray of ~15 km with an incremental step for each station. The subarray generally includes 3 BB stations and 25 SP stations (Supplementary Fig. 4a). The chosen subarray size is a compromise between the number of earthquakes recorded at BB and SP stations, as well as the inter-station spacing of the BB and SP stations. We have tested subarray sizes ~5 km larger or smaller but obtained similar results. Within each subarray, the subsurface structure is parameterized as the number of CRF phases, the timing, amplitude, and slowness of each phase, in which the slowness is used to link the subsurface structure beneath different stations within the subarray. We employ the McMC technique to iteratively add, remove, or adjust CRF phases, until the likelihood function (or misfit function) is stabilized[28]. Compared to the traditional time-domain iterative deconvolution, in which RF phases are fixed once added, McMC allows for dynamic updates of all phases throughout the process[28]. The resultant, more accurate constraint on earlier CRF phases (shallow structures) helps to improve the quality of the later phases (deeper structures). For each subarray, we run McMC Bayesian inversion using 500 CPUs to sample the model space simultaneously and independently. A total of 10,000 iterations are performed on each CPU. The first 5000 iterations are regarded as the "burn-in" period to allow convergence (Supplementary Fig. 4b), and every tenth model in the subsequent 5000 iterations is used to generate the ensemble solutions (Supplementary Fig. 4b). To present the CRF in a similar manner to the conventional RF, we collect all the CRF signals with the posterior probability over a prescribed threshold (e.g., larger than one standard deviation of the mean, that is 68% confidence

level) in the density map to construct the 1D trace that contains discrete delta functions with different timings, amplitudes, and slowness (Supplementary Fig. 4b). We then convolve the 1D trace with different Gaussian low-pass filters (Gaussian parameter a = 1.0, 2.0, 3.0, and 4.0) in each frequency band to generate the 1D CRF time series. We stack the 1D CRF waveforms (Supplementary Fig. 4c) from each subarray to produce the 2D stacking images (Supplementary Fig. 4d) via a slow common conversion point (CCP) stacking technique. Furthermore, we extract the posterior probability for the timing of CRF phases at each single-node subarray (Supplementary Fig. 4b), and we then stack the posterior probability distribution of the CRF phases to generate the 2D confidence images. Supplementary Fig. 6 shows the CRF imaging results obtained at multiple frequency bands. We also conduct CRF imaging using BB or SP data alone. Despite very different observation periods and instrument types, the CRF results constructed using the BB and SP data show remarkable agreement (Supplementary Fig. 6).

### Time-to-depth conversion using velocity models
To perform the time-to-depth conversion by accounting for the effect of the 3D velocity structure, we use the Vs tomography model from ref. 31 for time-to-depth conversion. The corresponding P-wave velocity and density are obtained through empirical relations[56]. However, using this model, as well as other global or regional tomographic models, in time-to-depth conversion results in an unrealistic thick slab-top low-velocity layer (LVL) with thickness >10 km and a relatively thin slab mantle of ~20 km (Supplementary Fig. 9), as explained in the main text. To better constrain the thickness of the slab-top LVL and the slab mantle, we pick arrivals related to the slab-top interface, slab-Moho, and slab-LAB, and conducted synthetic tests to understand the effect of velocity models in time-to-depth conversion (Supplementary Fig. 10). The observed arrival time difference between the slab-top interface and slab-Moho is about $1.3 \pm 0.35$ s. Following previous studies in Cascadia, assuming a fluid-saturated LVL with a very low Vs (about 20–50% Vs reduction) and a high Vp/Vs ratio (about 2–3) reduces the slab-top LVL thickness to ~7–8 km. Similarly, the observed arrival time difference between the slab-Moho and slab-LAB is $2.3 \pm 0.45$ s. The thickness of the slab mantle is thickened to ~25 km or larger by considering a 5–10% high-velocity subducting slab mantle and a normal Vp/Vs ratio (~1.75).

### Synthetic tests
To verify the robustness of our results presented in the main text, we test the effects of shallow structures' multiple reverberations on slab-LAB imaging in Supplementary Figs. 12–16. The synthetic seismograms are calculated with the Thomson–Haskell matrix method[57,58] using a 1D layered velocity model for each station that is extracted from the 2D velocity model. These synthetics are spatially arranged to mimic the effect of the 2D slab. Given the purpose of testing the effects of shallow multiples, not directly calculating 2D synthetic seismograms reduces the computational cost. Our previous studies have demonstrated the reliability of this approach[27,28]. We conduct five types of synthetic tests. Model-1 consists of a 7 km thick low-velocity subducting oceanic crust and a 33 km thick high-velocity subducting oceanic mantle, where the transition from the subducting lithosphere to the underlying asthenosphere is smooth. The slab geometry is obtained from the Slab2.0 model[59] in central Cascadia to mimic the real situations (Supplementary Fig. 12). Model-2 largely follows the structural features in Model-1, but consists of a sharp velocity reduction at the base of the subducting slab (Supplementary Fig. 12). By comparing the synthetics obtained from Model-1 and Model-2, we conclude that the imaged strong negative slab-LAB signal cannot be an imaging artifact due to the multiples from the slab-top and slab-Moho (Supplementary Fig. 12). In Model-3, we test the effects of basin structures on slab-LAB imaging, concluding that the basin reverberations occur at shallower depths than the imaged slab-LAB (Supplementary Fig. 12). In Model-4 and Model-5, we further

explored the effects of an intra-crustal discontinuity on slab-LAB imaging (Supplementary Fig. 13). Model-4 contains a positive Conrad discontinuity in the overriding continental plate at 15 km depth, while the Model-5 examines the negative intra-crustal discontinuity (at 10–15 km depth and at 60–130 km distance from coast) imaged in our study (Supplementary Fig. 13). In Model-4, the multiples of the Conrad discontinuity have similar arrivals as the direct Ps conversions of the slab-LAB. However, if the negative phase at around 8 s is the multiples of the Conrad discontinuity, we would expect to observe a similar magnitude positive phase just before the negative phase. In addition, the multiples from the Conrad discontinuity show different amplitude variations with respect to the incident directions (Supplementary Fig. 15), and the arrival time of the negative phase decreases with the increasing of epicentral distance (Supplementary Fig. 16). In Model-5, the multiples of the negative intra-crustal discontinuity exhibit different geometries compared to the imaged slab-LAB.

We further test the effect of an anisotropy layer on RFs using the synthetic data (Supplementary Figs. 17–19). The synthetic RFs are calculated using the Raysum code[54], which is a ray-based method to calculate synthetic seismograms for dipping discontinuities and anisotropic media. Analysis of observed RFs as a function of back-azimuths suggests that a dipping discontinuity provides a more plausible explanation for the negative signal observed at ~8 s.

### Multi-frequency receiver function waveform modeling
In this section, we conduct multiple frequency RF analyses to constrain the seismic structural characteristics related to the discontinuities observed in the main text. The amplitude of a P-to-S converted phase in RFs is sensitive to the S-wave velocity contrast across the discontinuity (dVs), the depth range over which the velocity contrast occurs (dZ), and the range of dominant frequencies used in analyses[29] (Supplementary Fig. 20). In this study, we only focus on the discontinuities observed in the distance range of 30–60 km to constrain their seismic characteristics, as they are less affected by the shallow basin structures (Supplementary Fig. 6). We pick the amplitudes related to the slab-top, slab-Moho, and slab-LAB discontinuities shown in the CRF images obtained using the BB data in Supplementary Fig. 6, as the SP data are limited by a narrower frequency band. We then constrain the dVs and dZ through a grid search inversion, in which the misfit is defined as the difference between the observations and synthetics,

$$\text{misfit} = \sum_{i=1}^{4} \left( \frac{\text{obs}_i - \text{syn}_i}{\text{obs}_i} \right)^2, \tag{1}$$

where the $i$ represents the observation or synthetic at a given frequency range.

For a given velocity model, the synthetic RFs are calculated with the Thomson–Haskell matrix method[57,58]. We generate a group of synthetic RFs with different bandpass filters similar to that used for real observations. We then pick the amplitudes of RF waveforms associated with each discontinuity and correct the amplitude by assuming the slab geometry from the Slab2.0 model[59]. In this region, the slab's average dip angle and strike direction are 356° and 12°, respectively, resulting in an amplitude correction factor of $1.3 \pm 0.1$ based on synthetic tests calculated using Raysum[54]. Supplementary Fig. 21 shows the relative misfit as a function of dVs and dZ. The relative misfit is defined as:

$$\text{Relative Misfit} = \frac{\text{Misfit}_i - \text{Misfit}_{\text{min}}}{\text{Misfit}_{\text{min}}} \times 100\%, \tag{2}$$

where the $\text{Misfit}_i$ represents the misfit of the $i$th model and $\text{Misfit}_{\text{min}}$ represents the minimum misfit in the grid-search inversion. The velocity structures for the slab-top and slab-LAB are well constrained, as we can fit the multiple frequency RF waveforms simultaneously.

The slab-Moho is not well constrained partially due to that the analysis of the Ps phase at low frequencies does not permit to distinguish the slab-Moho from the broader gradient of the high-velocity subducting slab mantle.

## Data availability

The seismic data used in this study are archived by the Incorporated Research Institutions for Seismology Data Management Center (IRIS DMC, network name: XZ_1993 and ZO_2017). All the seismic data used in this study are publicly available from the IRIS DMC. The subsurface structures obtained in this study, along with the scripts used to generate the figures, can be downloaded at https://doi.org/10.6084/m9.figshare.25434769.v1.

## Code availability

The codes used in this study are available to interested researchers upon request. Requests for material should be addressed to Ling Chen (lchen@mail.iggcas.ac.cn) or Xin Wang (wangxin@mail.iggcas.ac.cn).

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

## Acknowledgements

We thank the Incorporated Research Institutions for Seismology Data Management Center for providing seismological data. This study was supported by the National Key R&D Program of China grant 2022YFF0802600 (X.W.), National Natural Science Foundation of China grants 91958209 (Q-F.C.) and 42288201 (L.C.), National Science Foundation grant 1722879 (Z.Z.), the Young Elite Scientists Sponsorship Program by CAST grant 2020QNRC001 (X.W.), the Key Research Program of the Institute of Geology and Geophysics, Chinese Academy of Sciences No. IGGCAS-201904 (X.W.).

## Author contributions

X.W. designed the project, conducted the seismic analyses. X.W., L.C., K.W., Q-F.C., Z.Z. and J.Y. contributed to the interpretation of results. X.W. drafted the paper with substantial input from K.W. and L.C. and additional input from all co-authors.

## Competing interests

The authors declare no competing interests.
