## [Peer Review File · Nature Communications]

Seismic evidence for melt-rich Lithosphere-Asthenosphere Boundary beneath young slab at CascadiaREVIEWER COMMENTS

Reviewer #1 (Remarks to the Author):

This manuscript by Wang et al. identifies a coherent negative phase in the receiver functions obtained from dense linear array of seismometers deployed at the Cascadia subduction zone. The authors attribute the negative phase to the lithosphere-asthenosphere boundary (LAB) at the based on the subducting slab. Due to its sharpness, the LAB is interpreted to be at the melt-rich condition. The manuscript is overall well-written, and its novelty is clear. However, the method has some points needed to be further clarified. The requested revisions have the potential to impact the results, and thus, I recommend categorizing these changes as 'major revisions'.

1. Probability distribution

The procedure for calculating the probability distribution needs to be described with greater clarity. From my perspective, the process involves two steps: firstly, constructing a 1-D probability distribution for pulse presence by marginalizing the posterior probability across time for each sub-array. Subsequently, this 1-D time series of probabilities is either stacked or averaged within each bin on the cross-section using the CCP stacking method. It is important to note that, during these processes, there is no requirement for 'normalizing' the probability.

However, the manuscript repeatedly mentions normalization at both stages of analysis. The initial reference to normalization is found in the color scale of Fig S4(b), where it states "Normalized PDF (%)". I am not sure how the normalization is carried out and why it is considered necessary. It should be emphasized that improper normalization has the potential to distort the intended meaning of the probability distribution.

In the same figure, there appears to be some ambiguity regarding the connection between the left wiggle (labeled "PDF of CRF timing") and the right wiggle (labeled "CRF (PDF>68%)"). Specifically, the height of the peak at ~5.5 seconds in the left wiggle seems lower than half of the peak's height at ~6.5 seconds. Considering that probabilities should not exceed 100%, the peak at 5.5 seconds should represent a probability lower than 50%. However, in the right wiggle, the 5.5-second peak appears as if it carries a significant probability (68%). This inconsistency requires clarification.

Furthermore, the manuscript mentions the use of normalization for imaging at Line 386, but its necessity remains unclear to me. It seems that you can obtain the averaged probability in the image domain by simply summing the probabilities and dividing by the number of piercing traces at each bin. Therefore, an explanation for why normalization is employed in this context would be beneficial.

2. Forward simulation methods

The manuscript extensively uses forward simulations to demonstrate the robustness of the results but lacks information what specific method is used for the forward simulations, except for Figure 4, where the Thomson-Haskell matrix method is employed. Such information is essential to evaluate the validity of the analysis. It is particularly important to understand to what extent the simulations take into account 3-D heterogeneity and multiple reflections. Some methods based on ray theory can easily overlook the effects of multiple reflection phases.

Related to this, it is unclear why Thomson-Haskell's method was chosen for Figure 4. This method is applicable to flat layered structures without dipping interfaces. Since you seem to use the method that allows dipping interfaces (e.g., Figure S15-18), it raises the question of why the same method was not used for the analysis of Figure 4. If there is a specific reason for using the Thomson-Haskell method, it would be helpful to elaborate on how you corrected for any dipping effects.

3. Iteration numbers

I have some concerns regarding the relatively small number of iterations, which is set at 10,000 in your analysis. This number is notably lower than what is typically used in Markov Chain Monte Carlo (MCMC) analyses. It would be beneficial to justify the adequacy of this iteration number, for example, by demonstrating the evolution of the likelihood over the course of iterations.

4. Other minor comments

4-1. Line 366: What is the size of the increment for the moving sub-array?

4-2. Figure 4: I think that not all results from the grid-search is displayed on these Figures

4(a, b, and c). How did you select the ones to be displayed?

4-3. Fig. S11: The negative phase interpreted as slab-LAB shows a dip almost zero. Do you have any justification on it?

Reviewer #2 (Remarks to the Author):

A review of

Seismic evidence for a melt-rich Lithosphere-Asthenosphere 1 Boundary along the base of young slab at Cascadia by Wang et al

This paper uses receiver functions to provide images of the LAB beneath the Cascadia margin. This a young subduction system so it is of interest to see if results from here compared with what is found in Japan and New Zealand (NZ) where the subducted plate is much older.

The study presents a compelling case for an image of the LAB. The authors present an exhaustive test of different options for the RF signal in terms of potential multiples and convincingly make their case. I think there are some 23 figures in the Supp material !

The synthetic modelling is especially good and relevant.

The observation that the depth to the LAB is similar to that seen in Japan and NZ is not what I would have expected given recent findings showing the LAB increasing its depth with age. However, we are still very much in the learning stage of the LAB and this is just an added new observation to the mix.

I was interested to see the authors argue boundary at the top of the LAB is sharp . There is a sentence that reads: Our inferred slab-LAB consistently exhibits an amplitude of $12\pm 3\%$ of the direct P-arrival across the entire frequency range (Fig. 4), indicating a sharp discontinuity with a dVs of $9.8\pm 1.5\%$ over a depth range of 1.5 ± 1.5 km (Figs. 4 and S21).

The first part of sentence does not make sense (a slab does not exhibit an amplitude !) and

in general the whole sentence needs some unpacking. Why not say what the wavelength is and then apply the wavelength length rule as given in lines 242 and 243.

Or maybe use the figure that Stern et al (2015) have in their supplementary material (Fig. S7f) that shows the relationship of Reflection coefficient Vs normalised thickness (i.e. true thickness of transition zone /wavelength). This is an important finding of your paper and needs more explanation.

Have the authors seen : 128. Herath, P., Stern, T. A., Savage, M. K., Bassett, D., and Henrys, S., 2022, Wide-angle seismic reflections reveal a lithosphere-asthenosphere boundary zone in the subducting Pacific Plate, New Zealand: Science Advances, v. 8, no. 38, p. eabn5697.

In this paper it is argued that the very top of the plate is a thin layer of very high P-wave speed, due to azimuthal anisotropy, an immediately beneath it there is the low speed region. Are your results consistent with this ? or not ?

Editorial issues:

Line 42: I would suggest you remove the word “very”. It is meaningless in a science context, and the sentence becomes stronger without that word. But your choice.

Line 55 A note about the origin of the idea of a thin, low-velocity, melt layer at the base of tectonics plates facilitating plate motion and allowing plate tectonics to work. In your paper you ascribe this idea to Debayle et al (2020). My sense was that the first group to make the discovery of this thin layer, and point out that it is this layer that facilitates plates to slide, was the Stern et al (2015) paper. It would be appropriate to credit the first study that made this important link.

Line 82: another “very” that could be removed.

Line 269 : I think the word “at” should be “of” .

In summary this is an important new finding and should be published. The paper is well

presented . Figures are exceptional and it is well written.

My recommendation that minor edits, as listed above, should be made then it is ready to go.

Reviewer #3 (Remarks to the Author):

Review Wang, Chen et al 2024

This manuscript presents a new high resolution image of the LAB of subducting plate beneath Cascadia. Interpretation of the phase as the LAB seems sound given that its depth agrees with the high resolution tomography in the region. Similarly, much work has been done to demonstrate the robustness of the interpretation (SI).

The study is novel in that it is high resolution, requiring a sharp discontinuity at the base of the plate and shows that this is occurring down dip of locations where the LAB has by imaged by methods that do not resolve dipping structures as well. This is important to our understanding of both the LAB and subduction dynamics.

The authors interpretation of melt is likely sound. However, it would be nice to see a little more logic to back it up.

There has been a lot of talk in the community that a new paper (Liu et al) shows hydration can explain seismic observations and melt is not needed. However, in reality, overall, the effects of hydration on seismic waves are mixed depending on study – not just the Liu paper (see Cline, Faul, et al), and it remains to be seen which study is correct and why. In addition, the Liu result does not explain much more than a percent or of velocity reduction at shorter periods like those of P-to-S assuming mild hydration (150 ppm), and larger percentages of hydration likely cause melt at asthenospheric P/T conditions. That is another reason that this study (using higher freq P-to-S) is important in the debate on the LAB.

In terms of whether melt affects the viscosity of the mantle and therefore defines the LAB:

Lab experiments diverge on effect of small amount of melt on viscosity (Hirth & Kohlstedt, 1995; Mei & Kohlstedt, 2000)(Faul & Jackson, 2007). Theoretical models have argued that very small fractions (< 0.01 %) would have very little effect (Rudge, 2018) or up to 2 orders of magnitude of reduction (Holtzman, 2016). However, at higher melt fractions such as those required by this study (>1 %), there is likely a large effect on viscosity. Larger melt fractions have been reported by several observationally based papers, but there is still disbelief by many in the community that such percentages could be stable in the mantle over time and length scales to be imaged seismically. This is another aspect of the result that is novel – i.e., this is a big deal, something that needs more work. BTW, some 2 phase flow models (Katz, Joyce Sim, Turner, etc.) do find melt can pond beneath a permeability boundary in large percentages. So, again a hot topic of debate and something that needs to be addressed.

What is the interpretation of the negative intracrustal discontinuity of the upper plate?

I suggest breaking this into 2 sentences and explaining a little more.

“The reduction in the amplitude of the Ps conversion from the slab top 188 at ~40 km depth, as well as the absence of continental Moho near the mantle wedge (Fig. 3), is 189 consistent with the diminished velocity contrast between the hydrated forearc mantle wedge and 190 the subducting or overlying continental crust (Bostock et al., 2002; Kawakatsu & Watada, 2007).”

Also, the lack of a slab Moho sometimes interpreted as eclogitization of basaltic crust at least in the Bostock work.

“The denser spatial coverage of the short-period array allows us to better constrain shallow structures which aids in constraining deep structures.” What shallow structures? Can you be more specific? Also, how so? I thought that you were using an independent velocity model to translate to depth. How does information about shallower structures improve deeper structures?

“Recent geophysical studies suggest its persistence with down-going old and cold slabs, but it is not known whether it is commonly present, especially for young and warm slabs such as in the Cascadia subduction zone.” This is not true. Several authors infer melt beneath the slab in Cascadia (e.g Hawley). Please rewrite for accuracy.

“Nonetheless, 277 neither of these tomographic studies has adequate spatial resolution to detect the melt-rich LAB imaged in our study.” This is vague. Are you sure seismic tomography would not detect it, even if blurred by resolution? Please be more specific.

“We speculate that the melts at both the plate- and slab-LAB have a similar local origin, without 280 long-distance lateral migration.” This statement and paragraph need more logic. How/why does the observation indicates the melt has not migrated? In general, seismology provides a snapshot of the mantle and has difficulty deciphering the past/any kind of time differential. Maybe consider results from geodynamic modeling?

Line 165: Fig S11 does not look comparable to Fig. S3, instead the LAB seems to be flat and approaching the Moho toward the right of the figure. Please explain. Is this predicted for raypaths from NW?

References

Sim, S., 2018. The depth of mid-ocean ridges through Earth’s evolution and a two-phase study of melt focusing at mid-ocean ridges, Scripps Institution of Oceanography. University of California San Diego, San Diego, CA.

Cline, C.J., Faul, U.H., David, E.C., Berry, A.J., Jackson, I., 2018. Redox-influenced seismic properties of uppermantle olivine. *Nature* 555, 355-358 doi:10.1038/nature25764.

Liu, J., Tao, C., Zhou, J., Shimizu, K., Li, W., Liang, J., Liao, S., Kuritani, T., Deloule, E., Ushikubo, T., Nakagawa, M., Yang, W., Zhang, G., Liu, Y., Zhu, C., Sun, H., Zhou, J., 2022. Water enrichment in the mid-ocean ridge by recycling of mantle wedge residue. *Earth Planet Sc Lett* 584, 117455 doi:<https://doi.org/10.1016/j.epsl.2022.117455>.

Authors' Response to the Reviewers' Comments

Contents:

Response to Comments from Reviewer #1	1
Response to Comments from Reviewer #2	7
Response to Comments from Reviewer #3	10

Notes:

- Reviewers' comments are shown in black, and our response are shown in blue.
- The line numbers in this letter correspond to *Slab-LAB-maintext-with-changes-tracked.pdf*.
- Fig. x ----- Figures in the revised main text.
- Fig. Sx ----- Figures in the revised supplementary material.
- Fig. Rx ----- Figures in the response letter.

Response to Comments from Reviewer #1

General comments:

This manuscript by Wang et al. identifies a coherent negative phase in the receiver functions obtained from dense linear array of seismometers deployed at the Cascadia subduction zone. The authors attribute the negative phase to the lithosphere-asthenosphere boundary (LAB) at the based on the subducting slab. Due to its sharpness, the LAB is interpreted to be at the melt-rich condition. The manuscript is overall well-written, and its novelty is clear. However, the method

has some points needed to be further clarified. The requested revisions have the potential to impact the results, and thus, I recommend categorizing these changes as 'major revisions'.

Response: We appreciate the reviewer's great efforts to review the paper and provide valuable suggestions. Please find detailed point-by-point replies to all the comments below.

Comment 1: Probability distribution

The procedure for calculating the probability distribution needs to be described with greater clarity. From my perspective, the process involves two steps: firstly, constructing a 1-D probability distribution for pulse presence by marginalizing the posterior probability across time for each sub-array. Subsequently, this 1-D time series of probabilities is either stacked or averaged within each bin on the cross-section using the CCP stacking method. It is important to note that, during these processes, there is no requirement for 'normalizing' the probability.

However, the manuscript repeatedly mentions normalization at both stages of analysis. The initial reference to normalization is found in the color scale of Fig S4(b), where it states "Normalized PDF (%)". I am not sure how the normalization is carried out and why it is considered necessary. It should be emphasized that improper normalization has the potential to distort the intended meaning of the probability distribution.

In the same figure, there appears to be some ambiguity regarding the connection between the left wiggle (labeled "PDF of CRF timing") and the right wiggle (labeled "CRF (PDF>68%)"). Specifically, the height of the peak at ~5.5 seconds in the left wiggle seems lower than half of the peak's height at ~6.5 seconds. Considering that probabilities should not exceed 100%, the peak at 5.5 seconds should represent a probability lower than 50%. However, in the right wiggle, the 5.5-second peak appears as if it carries a significant probability (68%). This inconsistency requires clarification.

Furthermore, the manuscript mentions the use of normalization for imaging at Line 386, but its necessity remains unclear to me. It seems that you can obtain the averaged probability in the image domain by simply summing the probabilities and dividing by the number of piercing traces at each bin. Therefore, an explanation for why normalization is employed in this context would be beneficial.

Response: Thank you for your comments.

Regarding the normalization process shown in Fig. S4b, our intention was to show the ensemble solutions in terms of their density distribution as a function of timing and slowness of the Coherent Receiver Functions (CRF). In our Markov chain Monte Carlo (McMC) inversion, the ensemble solutions are initially represented by the counts of the accepted models (the CRFs). For visualization, we convert the counts into a density distribution, scaling the maximum value to 1, and we use the density plot to represent the Posterior Probability Distribution (PDF) of the CRFs. This approach was chosen to standardize the visual comparison across different subarrays. In the revised version, we have removed the term "normalization" and updated the labels to avoid any confusion, acknowledging that the use of the density plot to represent PDF is a common practice in McMC studies. Please see revised Fig. S4b.

Regarding the inconsistency between the left and right wiggles in Fig. S4b, we appreciate you bringing this to our attention. The right wiggle plot displays the 1D CRF with a 68% confidence level. To present the CRF in a manner similar to the conventional Receiver Function (RF) which contains discrete delta functions with different timings and amplitudes, we select the peaks with the probability over a certain threshold (e.g., larger than one or two standard deviations of the mean) in the 2D density map to construct the 1D CRF. We now realize that using "normalized probability" with "confidence level" could cause confusion, since both are expressed as percentage. To avoid confusion, we have revised Fig. S4b and its caption.

Comment 2: Forward simulation methods

The manuscript extensively uses forward simulations to demonstrate the robustness of the results but lacks information what specific method is used for the forward simulations, except for Figure 4, where the Thomson-Haskell matrix method is employed. Such information is essential to evaluate the validity of the analysis. It is particularly important to understand to what extent the simulations take into account 3-D heterogeneity and multiple reflections. Some methods based on ray theory can easily overlook the effects of multiple reflection phases.

Related to this, it is unclear why Thomson-Haskell's method was chosen for Figure 4. This method is applicable to flat layered structures without dipping interfaces. Since you seem to use the method that allows dipping interfaces (e.g., Figure S15-18), it raises the question of why the same method was not used for the analysis of Figure 4. If there is a specific reason for using the Thomson-Haskell method, it would be helpful to elaborate on how you corrected for any dipping effects.

Response: Thanks for the comments and suggestions.

For the synthetic studies shown in Figures S2 and S15-S18, we used the Raysum code (Frederiksen & Bostock, 2000), which is a ray-based method to calculate synthetic seismograms for dipping discontinuities and anisotropic media. For the 2D synthetic studies shown in S7 and S12-S14, we calculated the synthetic seismograms with the Thomson-Haskell matrix method using a 1D layered velocity model extracted from the 2D velocity model for each station. These synthetics are spatially arranged to mimic the effect of the 2D slab. Given the purpose of testing the effects of shallow multiples, not directly calculating 2D synthetic seismograms reduces computational cost. Our previous studies have demonstrated the reliability of this approach (Wang et al., 2021; Zhong & Zhan, 2020). These details have been added in the revised manuscript (lines 434-439 and lines 461-464).

In the multi-frequency RF waveform modeling section, we use the Thomson-Haskell's method for synthetic calculation. This method is chosen mainly for its computational efficiency, particularly when a larger number of layers is required to represent gradual transitions in velocity models. Although the Raysum code is an alternative with advantages in dealing with dipping layers, its application to a larger number of layers gives rise to extremely long run times and leads to segmentation faults (typically got segmentation fault with more than 8 layers, see Fig. 6 in Bloch and Audet (2023)). In using the Thomon-Hakell's method, we must account for the dipping effect. To address this, we use the slab's average dip (12°) and strike (356°) from the Slab2.0 model (Hayes et al., 2018) and estimate the amplitude correction factor ($\sim 1.3 \pm 0.1$) based on synthetic tests (Fig. R1). We have further clarified these points in the revised manuscript (lines 485-487).

Figure R1. The top panel shows the distribution of broadband telseismic waveforms used in the RF study across different back azimuth. The lower panel shows the variations in Ps amplitude ratio as a function of back-azimuth for a synthetic model with a dipping LAB interface, in comparison to a refence model with a flat LAB interface.

Comment 3: Iteration numbers

I have some concerns regarding the relatively small number of iterations, which is set at 10,000 in your analysis. This number is notably lower than what is typically used in Markov Chain Monte Carlo (MCMC) analyses. It would be beneficial to justify the adequacy of this iteration number, for example, by demonstrating the evolution of the likelihood over the course of iterations.

Response: Our Markov Chain Monte Carlo (MCMC) inversion was done using 500 CPUs, with each CPU running an independent chain. For each chain, 10,000 iterations were performed. Consequently, the total number of the models in the MCMC inversion are five million. Fig. R2a shows the evolution of the misfit (normalized to the “best” model) with iterations for the 500 parallel chains, and Fig. R2b shows the distribution of misfit for all ensemble solutions. From Fig. R2, we can see that the inversion typically convergences within only 5,000 iterations. We have updated Fig. S4 in the manuscript to show the evolution of the misfit over iterations.

Figure R2. Relative misfit (normalized to the “best” model) against iterations for 500 chains running in parallel. Each chain contains 10,000 MCMC iterations, with the second half of the iterations used to generate ensemble solutions. The histogram shown in the right panel shows the distribution of relative misfits for all the ensemble solutions.

Comment 4: Other minor comments

4-1. Line 366: What is the size of the increment for the moving sub-array?

Response: We used a moving sub-array of ~15 km with an incremental step for each station. See lines 382-383 in the revised manuscript.

4-2. Figure 4: I think that not all results from the grid-search is displayed on these Figures 4(a, b, and c). How did you select the ones to be displayed?

Response: We only displayed those models with relative misfit smaller than 100%. We have provided further clarification in the revised manuscript (lines 262-264 and lines 488-491).

4-3. Fig. S11: The negative phase interpreted as slab-LAB shows a dip almost zero. Do you have any justification on it?

Response: The slab-LAB imaged using waveforms from the northwest appears flat, and the thickness of the slab seems to vary along the down-dip direction. We have conducted justification on this feature by considering major factors that may affect it. The Bayesian posterior probability

distribution of the CRFs shows a high confidence level for this discontinuity (Fig. S11b), suggesting that noise effects may not be significant on the image. The observed slab-LAB roughly resembles the shape of the high-velocity slab shown in the tomographic result (Fig. S11c), although it manifests at a depth ~ 10 km deeper at distances of ~ 0 -40 km along the profile. This discrepancy is unlikely to result solely from the velocity model used in the time-to-depth conversion, as recognizing a 10 km decrease in the slab-LAB depth without changing the slab-Moho depth would require a slab mantle velocity $\sim 30\%$ lower than that of the current velocity model, which is unreasonable given the absolute velocities used (Fig. S11c). Further studies are required to understand the 3D slab effects in RF imaging and to incorporate more accurate tomographic models in the time-to-depth migration. If the flat appearance of the imaged slab-LAB shown in Fig. S11 reflects a real structural feature, it could suggest the presence of significant lateral variations of the slab structure. Indeed, both the strong lateral variation of slab geometry shown in the Slab 2.0 model (black dashed lines shown in Fig. S1) and strong variations of the plate-LAB depth observed offshore of Cascadia (Fig. S23 in the supplementary material) suggest a complex slab structure in this region. We have added some words about our analysis on the shape of the slab-LAB in the caption of Fig. S11.

Response to Comments from Reviewer #2

General comments:

This paper uses receiver functions to provide images of the LAB beneath the Cascadia margin. This a young subduction system so it is of interest to see if results from here compared with what is found in Japan and New Zealand (NZ) where the subducted plate is much older. The study presents a compelling case for an image of the LAB. The authors present an exhaustive test of different options for the RF signal in terms of potential multiples and convincingly make their case. I think there are some 23 figures in the Supp material!

The synthetic modelling is especially good and relevant.

The observation that the depth to the LAB is similar to that seen in Japan and NZ is not what I would have expected given recent findings showing the LAB increasing its depth with age. However, we are still very much in the learning stage of the LAB and this is just an added new observation to the mix.

I was interested to see the authors argue boundary at the top of the LAB is sharp. There is a sentence that reads: Our inferred slab-LAB consistently exhibits an amplitude of $12\pm 3\%$ of the direct P-arrival across the entire frequency range (Fig. 4), indicating a sharp discontinuity with a dVs of $9.8\pm 15\%$ over a depth range of 1.5 ± 1.5 km (Figs. 4 and S21).

The first part of sentence does not make sense (a slab does not exhibit an amplitude !) and in general the whole sentence needs some unpacking. Why not say what the wavelength is and then apply the wavelength length rule as given in lines 242 and 243.

Or maybe use the figure that Stern et al (2015) have in their supplementary material (Fig. S7f) that shows the relationship of Reflection coefficient Vs normalised thickness (i.e. true thickness of transition zone /wavelength). This is an important finding of your paper and needs more explanation.

Response: Thank you very much for your encouraging comments and suggestions. In the revised version, we have revised our wording regarding the “slab-LAB exhibits an amplitude”. In addition, we have incorporated the wavelength rule, as suggested, to estimate the upper bound thickness of this discontinuity. Please refer to lines 242-247 in the revised manuscript for these updates. By the way, our results indeed show that the LAB at Cascadia is shallower than those in Japan and New Zealand, consistent with recent findings showing the LAB increasing its depth with age.

Have the authors seen : 128. Herath, P., Stern, T. A., Savage, M. K., Bassett, D., and Henrys, S., 2022, Wide-angle seismic reflections reveal a lithosphere-asthenosphere boundary zone in the subducting Pacific Plate, New Zealand: Science Advances, v. 8, no. 38, p. eabn5697.

In this paper it is argued that the very top of the plate is a thin layer of very high P-wave speed, due to azimuthal anisotropy, an immediately beneath it there is the low speed region. Are your results consistent with this ? or not ?

Response: Thank you for the reference.

Herath et al. (2022) used active seismic reflection data to investigate the LAB structure in the Hikurangi subduction zone and identified a series of seismic reflections at depths associated with the LAB. They interpreted these reflections as the presence of a LAB zone consisting of a thin, high-velocity anisotropic layer atop a low-velocity, melt-rich layer.

In our study, we use passive seismic data and the RF technique which can resolve the overall variation of seismic velocity with depth, i.e., a single negative discontinuity corresponding to the

LAB. Given the inherent differences in resolution and sensitivity between active and passive seismic data, our study does not have the ability to resolve the detailed structure of the LAB zone as described by Herath et al.

Specific comments:

Editorial issues:

Line 42: I would suggest you remove the word “very”. It is meaningless in a science context, and the sentence becomes stronger without that word. But your choice.

Response: Here, we feel that the word “very” is a useful part of the scientific discussion because it emphasizes that the observed 3 km thickness is unusually small.

Line 55 A note about the origin of the idea of a thin, low-velocity, melt layer at the base of tectonics plates facilitating plate motion and allowing plate tectonics to work. In your paper you ascribe this idea to Debayle et al (2020). My sense was that the first group to make the discovery of this thin layer, and point out that it is this layer that facilitates plates to slide, was the Stern et al (2015) paper. It would be appropriate to credit the first study that made this important link.

Response: Thanks for pointing out this problem. The reference has been added.

Line 82: another “very” that could be removed.

Response: The word “very” has been removed.

Line 269 : I think the word “at” should be “of”.

Response: We have rephrased the sentence (line 272).

In summary this is an important new finding and should be published. The paper is well presented . Figures are exceptional and it is well written.

My recommendation that minor edits, as listed above, should be made then it is ready to go.

Response: Thank you very much for your encouraging comments.

Response to Comments from Reviewer #3

General comments:

This manuscript presents a new high resolution image of the LAB of subducting plate beneath Cascadia. Interpretation of the phase as the LAB seems sound given that its depth agrees with the high resolution tomography in the region. Similarly, much work has been done to demonstrate the robustness of the interpretation (SI).

The study is novel in that it is high resolution, requiring a sharp discontinuity at the base of the plate and shows that this is occurring down dip of locations where the LAB has been imaged by methods that do not resolve dipping structures as well. This is important to our understanding of both the LAB and subduction dynamics.

The authors interpretation of melt is likely sound. However, it would be nice to see a little more logic to back it up.

There has been a lot of talk in the community that a new paper (Liu et al) shows hydration can explain seismic observations and melt is not needed. However, in reality, overall, the effects of hydration on seismic waves are mixed depending on study – not just the Liu paper (see Cline, Faul, et al), and it remains to be seen which study is correct and why. In addition, the Liu result does not explain much more than a percent or of velocity reduction at shorter periods like those of P-to-S assuming mild hydration (150 ppm), and larger percentages of hydration likely cause melt at asthenospheric P/T conditions. That is another reason that this study (using higher freq P-to-S) is important in the debate on the LAB.

In terms of whether melt affects the viscosity of the mantle and therefore defines the LAB: Lab experiments diverge on effect of small amount of melt on viscosity (Hirth & Kohlstedt, 1995; Mei & Kohlstedt, 2000)(Faul & Jackson, 2007). Theoretical models have argued that very small fractions (< 0.01 %) would have very little effect (Rudge, 2018) or up to 2 orders of magnitude of reduction (Holtzman, 2016). However, at higher melt fractions such as those required by this study (>1 %), there is likely a large effect on viscosity. Larger melt fractions have been reported by several observationally based papers, but there is still disbelief by many in the community that such percentages could be stable in the mantle over time and length scales to be imaged seismically. This is another aspect of the result that is novel – ie., this is a big deal, something that needs more work. BTW, some 2 phase flow models (Katz, Joyce Sim, Turner, etc.) do find

melt can pond beneath a permeability boundary in large percentages. So, again a hot topic of debate and something that needs to be addressed.

Response: Thanks for your insightful comments and constructive suggestions.

We agree with your assessment of the findings of Liu et al. (2023). Considering the specific frequencies (up to 2 Hz) utilized in our study, hydration alone may not explain the substantial velocity reductions observed at the slab-LAB. In the revised manuscript, we have added this perspective. Please see lines 250-255 in the revised manuscript.

Regarding the questions about how melt affects the viscosity and the stability of melts over geologically timescales, we agree with the reviewer and have added a few sentences in the revised manuscript. See lines 296-298 and lines 316-320 in the revised manuscript.

Specific comments:

Comment 1:

What is the interpretation of the negative intracrustal discontinuity of the upper plate?

Response: The negative intracrustal discontinuity may be associated with the accumulation of crustal fluids, resulting from the fluids released by subducting sediments or oceanic crust, or from the heating and dewatering of the mantle wedge, as observed by previous studies in Cascadia (Egbert et al., 2022; Wang et al., 2019; Ward et al., 2018) and other subduction zones (Worzewski et al., 2011). We have added a sentence (lines 169-172) to clarify this.

Comment 2:

I suggest breaking this into 2 sentences and explaining a little more.

“The reduction in the amplitude of the Ps conversion from the slab top at ~40 km depth, as well as the absence of continental Moho near the mantle wedge (Fig. 3), is consistent with the diminished velocity contrast between the hydrated forearc mantle wedge and the subducting or overlying continental crust (Bostock et al., 2002; Kawakatsu & Watada, 2007).” Also, the lack of a slab Moho sometimes interpreted as eclogitization of basaltic crust at least in the Bostock work.

Response: Thank you for your suggestions. We agree that the original sentence does not make it clear that the mechanisms for the absence of the slab top at ~40 km depth and the absence of continental Moho are different. The absence of the slab top at ~40 km depth may primarily be attributed to the eclogitization of the subducting oceanic crust, as discussed by Bostock et al.

(2002) and Rondenay et al. (2008). With the progressive eclogitization of the oceanic crust, the slab Moho also becomes invisible as anhydrous eclogite cannot be distinguished from the underlying slab mantle on the basis of seismic wave speeds (Bostock et al., 2002; Chen et al., 2005). We have refined our sentences as follows (Lines 186-190): “The decrease in the amplitude of the Ps conversions at the slab top and slab-Moho when slab depth exceeds ~40 km (Fig. 3) is likely due to the eclogitization process of the subducting oceanic crust (Bostock et al., 2002; Rondenay et al., 2008). Eclogitization also releases aqueous fluid into the overlying forearc mantle wedge, leading to the well-known absence of a detectable continental Moho near the mantle wedge corner.”.

Comment 2:

“The denser spatial coverage of the short-period array allows us to better constrain shallow structures which aids in constraining deep structures.” What shallow structures? Can you be more specific? Also, how so? I thought that you were using an independent velocity model to translate to depth. How does information about shallower structures improve deeper structures?

Response: We appreciate the opportunity to provide further clarification on this matter. It's important to note that the procedures described does not involve time-to-depth conversion, but rather relates to the CRF method used in our study. Specifically, we employ the MCMC technique to iteratively add, remove, or adjust RF phases, until the likelihood function (or misfit function) stabilized. Compared to the traditional time-domain iterative deconvolution, in which RF phases are fixed once added, MCMC allows for dynamic update of all phases throughout the process. The resultant more accurate constraint on earlier CRF phases (shallow structures) helps to improve the quality of the later phases (deeper structures). The denser spatial coverage of the short-period array significantly improves our ability to constrain the shallow structures, such as basin structures. This will directly contribute to the deeper structures imaging (later CRF phases) in the CRF inversion. In the revised version, we have revised our statements to make it clearer (line 352 and lines 389-394).

Comment 3:

“Recent geophysical studies suggest its persistence with down-going old and cold slabs, but it is not known whether it is commonly present, especially for young and warm slabs such as in the

Cascadia subduction zone.” This is not true. Several authors infer melt beneath the slab in Cascadia (e.g Hawley). Please rewrite for accuracy.

Response: Hawley et al. (2016) reported a subslab low-velocity zone (LVZ) through tomography study along the entire Cascadia forearc and proposed that the LVZ reflects the accumulation of partial melts that were present as a thin layer beneath the slab. But in a later study by Bodmer et al. (2018), their tomography results show that the LVZs only exist in northern and southern Cascadia and proposed that they reflect partial melting due to localized mantle upwelling as opposed to a thin layer of partial melts below the slab. Thus, whether there is a thin layer of partial melts below the slab in Cascadia remains debated. In light of this, we have revised our statement (line 35) to more accurately reflect the current state of knowledge of the sub-slab partial melts below slab in Cascadia.

Comment 4:

“Nonetheless, neither of these tomographic studies has adequate spatial resolution to detect the melt-rich LAB imaged in our study.” This is vague. Are you sure seismic tomography would not detect it, even if blurred by resolution? Please be more specific.

Response: In our revised manuscript, we emphasize that these tomography models do not have enough vertical resolution to detect the sharp slab-LAB seismic discontinuity discussed in our study. We acknowledge that tomographic imaging may also be sensitive to the LVZs caused by partial melts, but its low vertical resolution prevents it from distinguishing whether the seismic velocity sharply or gradually changes with depth. In the revised manuscript, we have revised our statement to make it clearer (lines 287-289).

Comment 5:

“We speculate that the melts at both the plate- and slab-LAB have a similar local origin, without long-distance lateral migration.” This statement and paragraph need more logic. How/why does the observation indicate the melt has not migrated? In general, seismology provides a snapshot of the mantle and has difficulty deciphering the past/any kind of time differential. Maybe consider results from geodynamic modeling?

Response: Thanks for the comments. We have refined this sentence to make it clearer: “Considering the inference of melt-rich LAB with comparable melt fractions beneath oceanic

plates both before and after subduction (Rychert et al., 2020; Kawakatsu et al., 2009; Stern et al., 2015; this study), we speculate that the melts originate predominantly locally, without long-distance lateral migration from ridges or plumes.”. Please see lines 290-293 in the revised manuscript. We acknowledge this is an important question require further geodynamic studies.

Comment 6:

Line 165: Fig S11 does not look comparable to Fig. S3, instead the LAB seems to be flat and approaching the Moho toward the right of the figure. Please explain. Is this predicted for raypaths from NW?

Response: We think the reviewer meant Fig. 3 instead of Fig. S3 in this comment. A similar question has been raised by Reviewer #1. Please see our response to Reviewer #1’s Minor Comment 4-3 above.

Reference:

- Bloch, W., & Audet, P. (2023). PyRaysum: Software for Modeling Ray-theoretical Plane Body-wave Propagation in Dipping Anisotropic Media. *Seismica*, 2(1).
- Bodmer, M., Toomey, D. R., Hooft, E. E. E., & Schmandt, B. (2018). Buoyant Asthenosphere Beneath Cascadia Influences Megathrust Segmentation. *Geophysical Research Letters*, 45(14), 6954–6962. <https://doi.org/10.1029/2018GL078700>
- Bostock, M. G., Hyndman, R. D., Rondenay, S., & Peacock, S. M. (2002). An inverted continental Moho and serpentinization of the forearc mantle. *Nature*, 417(6888), 536–538. <https://doi.org/10.1038/417536a>
- Chen, L., Wen, L., & Zheng, T. (2005). A wave equation migration method for receiver function imaging: 2. Application to the Japan subduction zone. *Journal of Geophysical Research: Solid Earth*, 110(B11).
- Egbert, G. D., Yang, B., Bedrosian, P. A., Key, K., Livelybrooks, D. W., Schultz, A., et al. (2022). Fluid transport and storage in the Cascadia forearc influenced by overriding plate lithology. *Nature Geoscience*, 15(8), 677–682. <https://doi.org/10.1038/s41561-022-00981-8>
- Frederiksen, A. W., & Bostock, M. G. (2000). Modelling teleseismic waves in dipping anisotropic structures. *Geophysical Journal International*, 141(2), 401–412. <https://doi.org/10.1046/j.1365-246x.2000.00090.x>
- Hawley, W. B., Allen, R. M., & Richards, M. A. (2016). Tomography reveals buoyant asthenosphere accumulating beneath the Juan de Fuca plate. *Science*, 353(6306), 1406. <https://doi.org/10.1126/science.aad8104>
- Hayes, G. P., Moore, G. L., Portner, D. E., Hearne, M., Flamme, H., Furtney, M., & Smoczyk, G. M. (2018). Slab2, a comprehensive subduction zone geometry model. *Science*, 362(6410), 58–61.

- Herath, P., Stern, T. A., Savage, M. K., Bassett, D., & Henrys, S. (2022). Wide-angle seismic reflections reveal a lithosphere-asthenosphere boundary zone in the subducting Pacific Plate, New Zealand. *Science Advances*, 8(38), eabn5697. <https://doi.org/10.1126/sciadv.abn5697>
- Liu, C., Yoshino, T., Yamazaki, D., Tsujino, N., Gomi, H., Sakurai, M., et al. (2023). Effect of water on seismic attenuation of the upper mantle: The origin of the sharp lithosphere–asthenosphere boundary. *Proceedings of the National Academy of Sciences*, 120(32), e2221770120.
- Rondenay, S., Abers, G. A., & van Keken, P. E. (2008). Seismic imaging of subduction zone metamorphism. *Geology*, 36(4), 275–278. <https://doi.org/10.1130/G24112A.1>
- Wang, X., Zhan, Z., Zhong, M., Persaud, P., & Clayton, R. W. (2021). Urban Basin Structure Imaging Based on Dense Arrays and Bayesian Array-Based Coherent Receiver Functions. *Journal of Geophysical Research: Solid Earth*, 126(9), e2021JB022279.
- Wang, Y., Lin, F.-C., & Ward, K. M. (2019). Ambient noise tomography across the Cascadia subduction zone using dense linear seismic arrays and double beamforming. *Geophysical Journal International*, 217(3), 1668–1680. <https://doi.org/10.1093/gji/ggz109>
- Ward, K., Lin, F., & Schmandt, B. (2018). High-Resolution Receiver Function Imaging Across the Cascadia Subduction Zone Using a Dense Nodal Array. *Geophysical Research Letters*, 45(22), 12–218.
- Worzewski, T., Jegen, M., Kopp, H., Brasse, H., & Taylor Castillo, W. (2011). Magnetotelluric image of the fluid cycle in the Costa Rican subduction zone. *Nature Geoscience*, 4(2), 108–111. <https://doi.org/10.1038/ngeo1041>
- Zhong, M., & Zhan, Z. (2020). An array-based receiver function deconvolution method: methodology and application. *Geophysical Journal International*, 222(1), 1–14.

REVIEWERS' COMMENTS

Reviewer #1 (Remarks to the Author):

I appreciate the authors' efforts in enhancing the quality of the manuscript. My requirements have been satisfactorily addressed, and I am confident that the manuscript is now prepared for publication.

Reviewer #3 (Remarks to the Author):

Thank you for responding to the requested edits. This is a nice manuscript, of ide interest to many, worthy of publication.